# MRI-Based Radiomics Nomogram for Predicting Prostate Cancer with Gray-Zone Prostate-Specific Antigen Levels to Reduce Unnecessary Biopsies

**DOI:** 10.3390/diagnostics12123005

**Published:** 2022-12-01

**Authors:** Li Zhang, Jing Zhang, Min Tang, Xiao-Yan Lei, Long-Chao Li

**Affiliations:** 1Department of MRI, Shaanxi Provincial People’s Hospital, Xi’an 710068, China; 2Institute of Medical Research, Northwestern Polytechnical University, Xi’an 710068, China

**Keywords:** radiomics, MRI, prostate cancer, biopsies

## Abstract

Objective: The aim of this study was to establish a predictive nomogram for predicting prostate cancer (PCa) in patients with gray-zone prostate-specific antigen (PSA) levels (4–10.0 ng/mL) based on radiomics and other traditional clinical parameters. Methods: In all, 274 patients with gray-zone PSA levels were included in this retrospective study. They were randomly divided into training and validation sets (n = 191 and 83, respectively). Data on the clinical risk factors related to PCa with gray-zone PSA levels (such as Prostate Imaging Reporting and Data System, version 2.1 [PI-RADS V2.1] category, age, prostate volume, and serum PSA level) were collected for all patients. Lesion volumes of interest (VOI) from T2-weighted imaging (T2WI) and apparent diffusion coefficient (ADC) imaging were annotated by two radiologists. The radiomics model, clinical model, and combined prediction model, which was presented on a nomogram by incorporating the radiomics signature and clinical and radiological risk factors for PCa, were developed using logistic regression. The area under the receiver operator characteristic (AUC-ROC) and decision, calibration curve were used to compare the three models for the diagnosis of PCa with gray-zone PSA levels. Results: The predictive nomogram (AUC: 0.953) incorporating the radiomics score and PI-RADS V2.1 category, age, and the radiomics model (AUC: 0.941) afforded much higher diagnostic efficacy than the clinical model (AUC: 0.866). The addition of the rad score could improve the discriminatory performance of the clinical model. The decision curve analysis indicated that the radiomics or combined model could be more beneficial compared to the clinical model for the prediction of PCa. The nomogram showed good agreement for detecting PCa with gray-zone PSA levels between prediction and histopathologic confirmation. Conclusion: The nomogram, which combined the radiomics score and PI-RADS V2.1 category and age, is an effective and non-invasive method for predicting PCa. Furthermore, as well as good calibration and is clinically useful, which could reduce unnecessary prostate biopsies in patients having PCa with gray-zone PSA levels.

## 1. Introduction

According to the American Cancer Society, prostate cancer (PCa) was the most common malignancy and the second leading cause of cancer death among men in 2021 [1]. In clinical practice, prostate-specific antigen (PSA) has been widely used in PCa screening, because it is a good diagnostic measure for identifying PCa [2]. However, gray-zone PSA levels (4–10.0 ng/mL) have poor sensitivity and specificity for PCa prediction, and biopsy results have been reported to be positive in only 11.8–25% of patients with gray-zone PSA levels [3,4]. This indicates that 75–88.2% of patients undergo unnecessary needle biopsies and are at risk of complications [5]. In addition, PSA levels are affected by factors such as the volume of the prostate, inflammation of the prostate, and BMI [6,7,8,9]. Therefore, prebiopsy prediction of PCa in patients with gray-zone PSA levels is a clinical challenge.

To reduce unnecessary biopsies, some indicators are widely used, especially in patients with PSA levels in the range of 4–10 ng/mL, such as age, prostate volume (PV), free/total PSA ratio (PSA f/t ratio), and PSA density (PSAD). However, which indicators are more suitable for patients having PCa with gray-zone PSA levels remains controversial, and no consensus has been reached [10,11].

In recent times, multiparametric magnetic resonance imaging (mp-MRI) based on Prostate Imaging Reporting and Data System (PI-RADS) has shown a moderate-to-good diagnostic performance in the prediction of PCa in patients with gray-zone PSA levels [12,13,14]. However, PI-RADS has some limitations, especially low specificity and poor inter-reader reproducibility [12,13,14,15]. Thus, it is necessary to develop a quantitative and objective diagnostic method to improve the performance of PI-RADS for predicting PCa in patients with gray-zone PSA levels.

Radiomics can offer large numbers of imaging features to quantify specific tumor attributes and phenotypes. These features can be used to glean valuable diagnostic or prognostic information [16,17,18]. In recent years, MRI-based radiomics has been found to be potentially beneficial in predicting PCa, thereby providing an objective method based on much more information than can be visually analyzed by radiologists [18,19,20]. To date, radiomics signatures and clinical factors have been analyzed in combination to develop a radiomics nomogram, which is widely used for cancer prediction, along with building a graphically depictive statistical predictive model that is individualized [21]. Thus, the study aimed to develop and validate a nomogram combining radiomics and conventional clinical factors to evaluate the diagnostic accuracy for PCa in patients with gray-zone PSA levels. We also compared whether the combination nomogram of these methods helps improve diagnostic efficiency.

## 2. Materials and Methods

This retrospective study was conducted according to the TRIPOD reporting checklist (available at http://dx.doi.org/10.21037/tau-21-49, accessed on 21 December 2021). The institutional review board approved this study, and the requirement of informed consent was waived.

### 2.1. Patients

Between December 2015 and October 2021, 478 consecutive patients with PSA levels of 4–10 ng/mL who underwent prostate mp-MRI in Shaanxi Provincial People’s Hospital were included in this study. The patients were selected based on the following criteria: (1) prebiopsy prostate mp-MRI acquired according to the PI-RADS criteria; (2) MRI examination performed before standardized prostatic biopsy and/or prostatectomy and pathologic confirmation of PCa, inflammation, or prostatic hyperplasia; (3) no previous any treatments including prostate endocrine therapy, radiation therapy, or surgery before the MRI examination. The exclusion criteria were as follows: (1) MRI quality was poor for diagnosis or segmentation and (2) a lack of clinical data to construct the prediction model. The flowchart for the study population enrollment is shown in Figure 1. They were randomly divided into training and validation sets (*n* = 191 and 83, respectively).

### 2.2. MRI Protocol

MRI was performed using the 3.0-T Ingenia or 3.0-T Ingenia CX MR system (Philips Healthcare, Best, the Netherlands) with an abdominal eight-channel surface phased array coil. The main sequences included sagittal T2-weighted imaging (T2WI), axial T2WI, coronal T2WI, axial T1-weighted imaging (T1WI), diffusion-weighted imaging (DWI) (b values of 0, 1000, and 2000 s/mm^2^), Apparent diffusion coefficient (ADC) maps, and dynamic contrast-enhanced (DCE)-MRI. Gadopentetate dimeglumine (Omniscan, GE Healthcare, Milwaukee, WI; Magnevist; Bayer Healthcare) was intravenously injected at a dose of 0.1 mmol/kg of body weight and rate of 2.5 mL/s by using an automatic injector (Guerbet). The details of MRI acquisition parameters, including the sequence, flip angle (degree), repetition time (TR)/echo time (TE), echo train length, matrix size, field of view, thickness, and b values, are summarized in Appendix A).

### 2.3. Reference Standard

All lesions were histopathologically evaluated based on a standard 12-core systematic transperineal ultrasound-guided prostate biopsy or surgical specimens (radical prostatectomy). Two experienced genitourinary pathologists recorded the location and Gleason score, with all disagreements being resolved by consensus. The area of systematic prostate biopsy was divided as follows: the prostate gland was divided into 3 sections (the basal region, the body, and the apex) from top to bottom, and each part was divided into the left and right regions. The right and left regions of the basal and body regions were divided into the inner and outer regions, and the whole prostate gland was divided into 10 areas. Each area was punctured with 1 needle, and 2 additional needles were used to puncture the suspicious area.

### 2.4. Clinical Data

Age, PV, serum PSA level (including total PSA [tPSA] and free PSA [fPSA]), f/t PSA, and PSAD were collected for the patients selected. PV was calculated as follows: anteroposterior diameter × vertical diameter × transverse diameter × 0.52. PSAD was calculated as total PSA/PV. Two radiologists (L.C.L. with 7 years of experience in prostate MRI, and J.Z. with 5 years of experience in prostate MRI) independently assigned each lesion according to the PI-RADS V2.1. criteria; they were blinded to all clinicopathological information. For the transition zone (TZ) lesion, T2WI remains the key sequence, with using of DWI features to upgrade atypical nodules to category 3 lesions for staging lesions. For peripheral zone (PZ) lesions, DWI remains the key sequence, with using DCE for DWI score 3 lesions. The radiologists independently assigned each lesion a score of 1–5 for T2WI, a score of 1–5 for DWI, “+” or “−” for DCE, and an overall PI-RADS assessment category according to PI-RADS v2.1. For patients with multiple lesions, only the largest lesion in each patient was scored. Then the final PI-PADS V2.1 score was discussed with a third radiologist (L.X.Y. with 15 years of experience in prostate MRI) during image interpretation.

### 2.5. Radiomic Analysis: Segmentation and Extraction

A radiologist (L.C.L.) blinded to the clinical information and histopathological results manually performed segmentation using ITK-SNAP (http://www.itksnap.org, accessed on 15 January 2022) software independently. Volumes of interest (VOI) delineated the visible lesion slice by slice based on T2WI and ADC map, and then, VOI labels were automatically generated.

When multiple lesions were found, only the dominant lesion (the largest lesion) was considered. Finally, the PI-RADS V2.1 reports, biopsy results, and radiologists’ segmentation were matched for each patient. The location and size of the lesions were determined as follows: (I) the detailed records of the prostate system punctures (injection site and depth) and pathological diagnostic results were used to determine the location and nature of the lesion; (II) the location described by pathology was matched to the corresponding lesion on the MRI images; and (III) combined with the PI-RADS V2.1 scoring diagnostic criteria, the range of lesions was determined.

The open-source software Pyradiomics (v. 3.6.5; https://www.python.org, accessed on 25 January 2022) was used to extract radiomics features. The synthetic minority oversampling technique (SMOTE) was used to generate a sample from the joint weighting of multiparametric features [22]. A total of 3146 features of four types (first-order statistics, texture features, shape features, and wavelet filter and Laplacian of Gaussian filter features) were extracted from axial ADC and T2WI sequences.

Feature selection involved four steps. First, another radiologist (J.Z.) was provided with the lesion location based on pathology randomly selected 30 cases to segment. Using the intra-class coefficient (ICC) to evaluate the inter- and intraobserver stability of the obtained radiomics features from the two VOI sets, features with inter-class coefficients and ICCs > 0.75 were selected for the following analysis. Second, redundant features were removed by one-way analysis of variance. Third, Pearson’s correlation coefficient analysis was performed sequentially. Fourth, the least absolute shrinkage and selection operator regression (lasso) method was applied to select the most significant features for predicting PCa with gray-zone PSA levels.

### 2.6. Model Construction

For the clinical model, these indexes, including age, volume, tPSA, fPSA, F/T PSA, PSAD, and the PI-RADS V2.1 score, were assessed by univariate logistic regression. The features identified as statistically significant with univariate logistic regression analysis were then analyzed with multiple logistic regression analysis to build a model to identify associations between clinical features and PCa in the PSA gray zone.

For the radiomics model, a logistic regression algorithm was applied to construct the model in the training set to obtain the optimal parameters, and the model was then assessed in the validation set. Then, a radiomics score (rad score) was calculated for each patient via a linear combination of selected features that were weighted by their respective coefficients. The number of patients with PCa was smaller than the number of non-PCa patients, and this sample imbalance would have an adverse impact on the performance of a classifier. Thus, we used SMOTE to generate a sample.

For the clinical radiomics model, a nomogram was constructed using the combined clinical and radiomics model with the coefficients of clinical factors and radiomics features by multivariate logistic regression. The C-index was calculated to assess the discrimination performance of the radiomics nomogram. A calibration curve was plotted to explore the predictive accuracy of the nomogram [23].

The training and validation test was applied for all three models.

### 2.7. Clinical Usefulness

Decision curve analysis (DCA) was used to investigate the clinical utility of the nomogram in both the training and validation sets [24].

### 2.8. Statistical Analysis

All statistical analyses were performed using SPSS (v. 21.0.0.0; https://www.ibm.com, accessed on 28 January 2022), and R (v. 4.1.2; http://www.r-project.org/, accessed on 29 January 2022) software. The clinical features between the PCa and non-PCa were compared by applying the *t*-test, the Chi-square test, or the Mann-Whitney U test, as appropriate.

Univariate and multiple analyses were performed using logistic regression analysis to select significant predictors of PCa in the PSA gray zone. Odds ratios and 95% CIs were determined. The AUC-ROC curve was used to evaluate the performance of each model. Accuracy, sensitivity, specificity, negative predictive value (NPV), and positive predictive value (PPV), positive likelihood ratio (LR+), negative likelihood ratio (LR−) were calculated according to the Youden index.

Further, we evaluated the incremental discrimination ability of the clinical model, radiomics model, and clinical-radiomics model using the DeLong test [25]. *p* < 0.05 indicated statistical significance. The entire workflow of this analysis is presented in Figure 2.

## 3. Results

### 3.1. Patient Characteristics

A total of 274 patients with PSA in the gray zone were included in this study, of which 90 patients had PCa and 184 patients had benign lesions. In assessments of the lesion origin, 75 lesions were found to have originated from the PZ, while 199 lesions were located in the TZ. Among them, 177 lesions were benign prostatic hyperplasia which were located in the TZ. A total of 64 (33.5%) of 191 patients in the training cohort and 26 (31.2%) of 83 patients in the validation cohort were diagnosed with PCa. As for the reference standard, 184 patients with benign lesions and 41 patients with PCa did not undergo radical prostatectomy, and the pathological results were determined by TRUS-guided biopsy. Other 49 patients with PCa did undergo radical prostatectomy with surgical specimens. Table 1 shows the characteristics of all patients.

### 3.2. Clinical Model

The univariate logistic regression analysis suggested that among clinical factors, patient age, PV, tPSA, PSAD, and PI-RADS V2.1 score were significant factors for predicting PCa in patients with PSA in the gray zone. The multiple logistic analysis showed that age and the PI-RADS V2.1 score were important factors that could be used as independent predictors. The outcomes of the univariate and multiple logistic regression analyses are presented in Table 2.

Finally, the logistic regression classifier was established according to age and the PI-RADS V2.1 score. The AUC, sensitivity, and specificity of the training set were 0.866 (0.783–0.950), 0.842 (0.724–0.916), and 0.846 (0.655–0.941), respectively, in the validation set (Table 3).

### 3.3. Radiomics Model

The interobserver ICCs of the selected features ranged from 0.772 to 1, and the intraobserver ICCs ranged from 0.751 to 1. Thus, 3146 features extracted from T2WI images combined with ADC mapping were screened for subsequent analyses. Finally, 25 radiomics features (Appendix A) by the lasso model were included for the T2WI images and ADC mapping (Appendix A and Figure 2).

Rad scores were obtained (Appendix A), and the rad score bar diagrams are shown in Figure 3. The bar diagrams demonstrated good discrimination performance of the rad score. The radiomics signature yielded an AUC, sensitivity, and specificity of 0.982 (0.964–0.999), 0.953 (0.865–0.985), and 0.961 (0.909–0.984), respectively, for discriminating between PCa and non-PCa groups among patients with PSA in the gray zone in the training cohort and 0.941 (0.888–0.995), 0.808 (0.613–0.918), 0.965 (0.870–0.991), respectively, in the validation cohort (Table 3). The values of AUC, sensitivity, specificity, and accuracy by the SMOTE method were similar, and are illustrated in Appendix A.

### 3.4. Nomogram Model

The results showed that age (*p* = 0.0013), PI-RADS V2.1 score (*p* < 0.001), and the rad score (*p* < 0.001) were significantly different between PCa and non-PCa patients with PSA in the gray zone. The clinical radiomics nomogram incorporating age, PI-RADS V2.1, and rad score on the basis of multivariate logistic regression are shown in Figure 4. The clinical radiomics nomogram yielded an AUC, sensitivity, specificity, and C-index of 0.953 (0.907–0.999), 0.885 (0.697–0.962), 0.930 (0.870–0.991), and 0.953, respectively, in the validation cohort. The nomogram showed the same diagnostic efficacy as the radiomics model. The AUC, accuracy, sensitivity, specificity, PPV, and NPV of the three models are listed in Table 3. A comparison of the ROC curves of these three models is shown in Figure 5.

### 3.5. Calibration

The calibration curves obtained after the application of the nomogram in the training and validation groups are shown in Figure 6. The nomogram showed good agreement between prediction and histopathologic confirmation for detecting PCa in PSA gray zone.

### 3.6. Clinical Use

The DCAs for the clinical and radiomics models and the nomogram are shown in Figure 7. The DCAs indicated that PCa prediction using the radiomics or combined models added more benefit than prediction using the clinical model.

## 4. Discussion

Our study sought to construct and validate a nomogram incorporating the radiomics signature with two preoperative clinic characteristics (PI-RADS V2.1 score and age). The radiomics model and the combined model showed stronger ability than the clinical model in discriminating between PCa and non-PCa in cases with PSA in the gray zone. Furthermore, the addition of the rad score to the clinical model could increase the rate of identification of PCa-negative cases by 8.4% in comparison with the clinical model for predicting PCa in cases with PSA in the gray zone. Radiomics signatures based on the SMOTE-balanced dataset achieved similar synthesized performance as the original dataset. DCA demonstrated good clinical usefulness of the nomogram and suggested that it facilitated preoperative individualized prediction of PCa and could be used to reliably rule out PCa in the gray zone, obviating the need for biopsies.

Some previous studies have used PI-RADS for PCa screening [26,27]. Wang et al. [26] showed that for TZ patients with PSA levels of 4–20 ng/mL, the PI-RADS V2.1 score detected PCA with an AUC of 0.889. Sun et al. [27] incorporated risk factors, including the PSA level and PI-RADS V2 score into a nomogram, which showed better PCa prediction performance (AUC = 0.876) than PI-RADS V2 only. Niu et al. [21] used patient age, PI-RADS V2 score, and adjusted PSAD to develop a nomogram by the logistic regression model and showed an AUC of 0.85 (0.79–0.90) for predicting PCa in the gray zone. Through univariate and multiple logistic analyses, our clinical model combined the PI-RADS V2.1 score and age and showed an AUC of 0.866 in the validation set, which was consistent with the results reported by Niu et al. [21] and Sun et al. [27]. However, considering the differences in clinical factors and case grouping in the above-mentioned studies, a more simple and accurate method for predicting PCa in the gray zone should be sought in future studies.

Benign prostate lesions include glandular structures and large intercellular spaces. However, PCa shows high cellularity and reduced extracellular spaces. These differences in histopathological features can be reflected by radiomics methods. Radiomics involves more quantitative and objective features of the tumor, which can overcome the interobserver variability of PI-RADS, and potentially yield useful predictive biomarkers of tumor heterogeneity that cannot be discerned via visual analysis [28,29]. Li et al. [30] reported the development of a radiomics model to predict PCa. Their findings showed that the radiomics model has high predictive efficiency with AUC values of 0.99 and 0.98 in both training and validation sets, which were higher than our results (AUC of 0.982 in the training set and 0.941 in the validation set). This may be due to the fact that the previous study included patients with no limitations on the PSA level. In fact, the PSA in the gray zone, which groups up to 80% of biopsies, was unnecessary [5]. Therefore, a better risk prediction method specific to these patients is needed [21,31].

A similar study by Qi et al. [14] developed a radiomics signature incorporating the age, PSAD, and the PI-RADS V2 score and yielded AUC values of 0.956 and 0.933 in the primary and validation cohorts, respectively. However, this cohort used the PI-RADS V2 score, which was published in 2014, and did not develop a nomogram model.

Nomograms are widely used to predict medical prognoses and outcomes by combining multiple risk factors, representing a clearer, more concise, and easier-to-understand approach [32,33]. The present study used the radiomics signature, age, PI-RADS V2.1 score (published in 2019), after univariate and multiple analyses to determine significant predictors, and constructed a radiomics clinical nomogram that achieved high performance for PCa identification in this study. The addition of the rad Score to the clinical model can improve the diagnostic efficiency and clinical net benefit in “gray zone” PCa diagnosis.

In our study, the calibration curve indicated high accuracy for individual predictions in the training and validation sets (Figure 6). The DCA in the current study showed that using radiomics nomograms to predict PCa with PSA in the gray zone added more benefit than either the treat-all scheme or no-treatment scheme. The prediction model showed more advantages in guiding physician decision-making.

Several limitations of this study require consideration. First, due to its retrospective design, the study inevitably showed sample selection bias. Second, this was a single-center study without external validation. In future studies, our model should be assessed and verified in more centers with larger samples. Third, some patients underwent a biopsy as the reference test, which resulted in the risk of mismatching some cancers. Nevertheless, a systematic prostate biopsy is the recommended technique for a prostate biopsy according to the NCCN guidelines. Additionally, we identified lesions by comparing biopsy templates with mp-MRI results, trying our best to reduce the deviation. On the other hand, the biopsy results might have been affected by the underestimation of the Gleason Score with the diagnosis of PCa. In the future, studies are needed with the length of specimens >12 mm or targeted biopsies to avoid this underestimation [34]. Fourth, our study did not separate the PZ and TZ lesions due to the small number of enrolled patients. Therefore, further investigations are needed to expand the sample of the population and evaluate PZ and TZ lesions separately in different groups. Fifth, radiomics signatures extracted from our MR images only contained T2WI, and ADC maps of patients, such as DCE may provide more information and improve performance. This still needs further investigation.

## 5. Conclusions

In conclusion, the nomogram, which combined the radiomics score with the PI-RADS V2.1 category and age, is an effective and noninvasive approach for predicting PCa, and showed good calibration and clinical usefulness, which could reduce unnecessary prostate biopsies in patients with PSA in the gray zone. The addition of the rad score could improve the discrimination performance of the clinical model. Radiomics characteristics are obviously valuable in diagnosing PCa in the PSA gray zone.

## Figures and Tables

**Figure 1 diagnostics-12-03005-f001:**
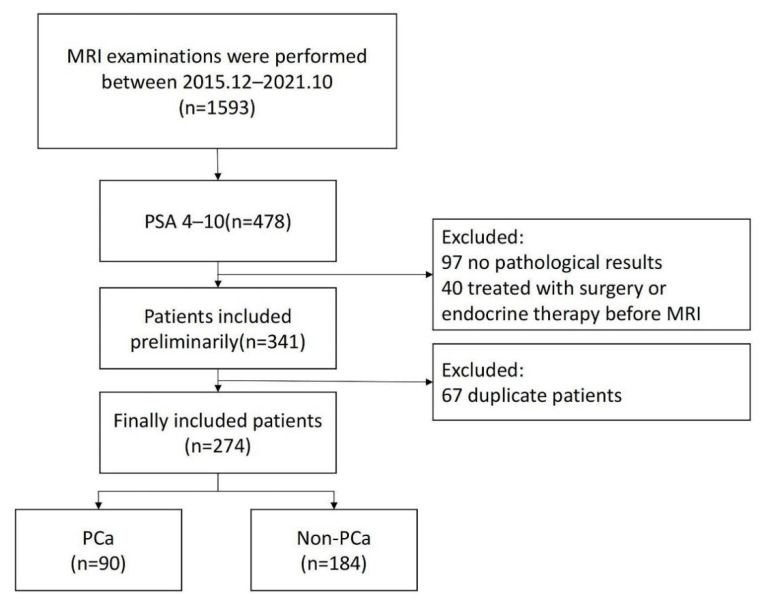
Patient selection flowchart.

**Figure 2 diagnostics-12-03005-f002:**
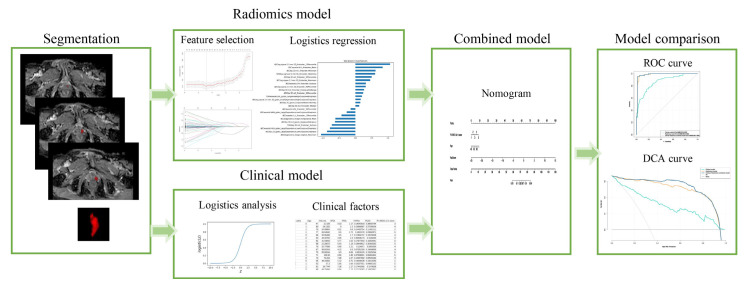
The workflow of this study.

**Figure 3 diagnostics-12-03005-f003:**
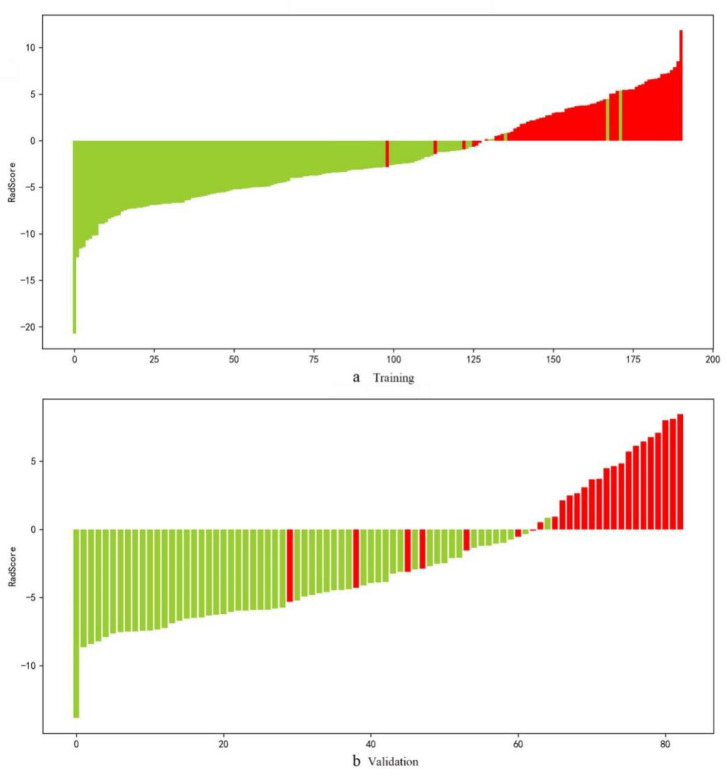
Bar diagrams in the training (**a**) and validation (**b**) groups. Up and down bars refer to the predicted malignant and benign lesions, respectively. Red and green bars refer to actual malignant and benign lesions, respectively.

**Figure 4 diagnostics-12-03005-f004:**
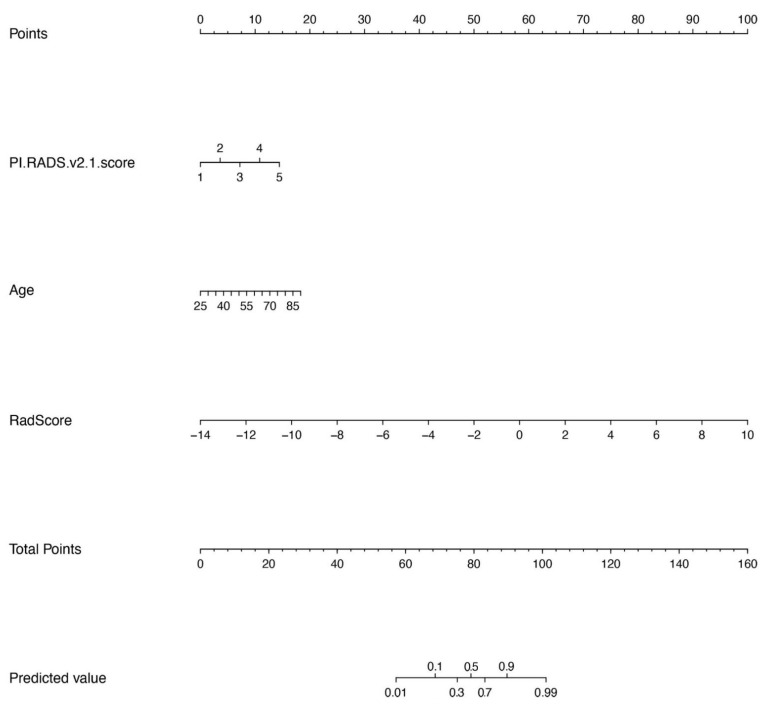
Nomogram with the radiomics score (Rad-score) and clinical factors incorporated.

**Figure 5 diagnostics-12-03005-f005:**
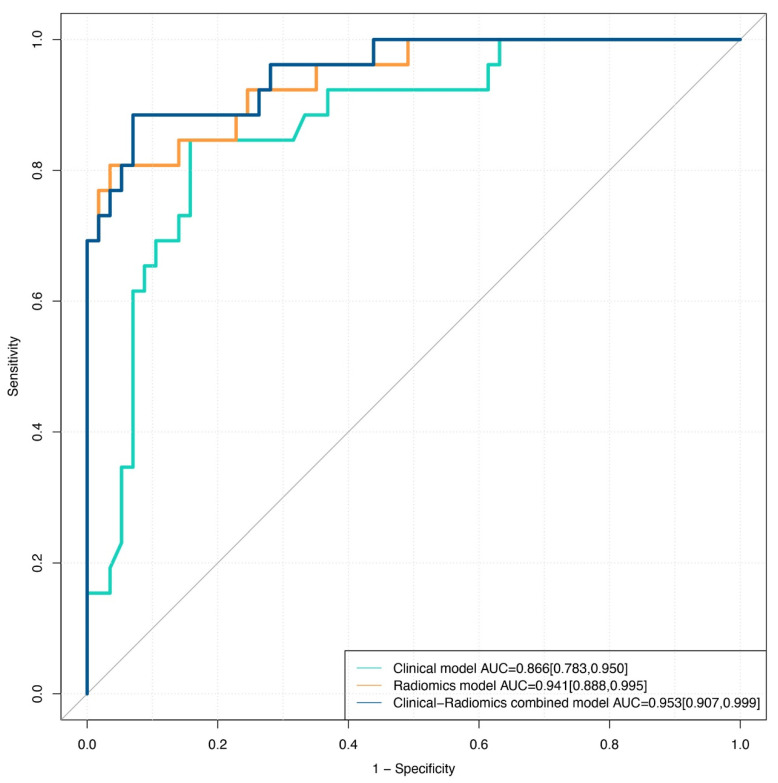
Comparison of ROC curves for differentiation of the three models for predicting PCa in the validation cohort.

**Figure 6 diagnostics-12-03005-f006:**
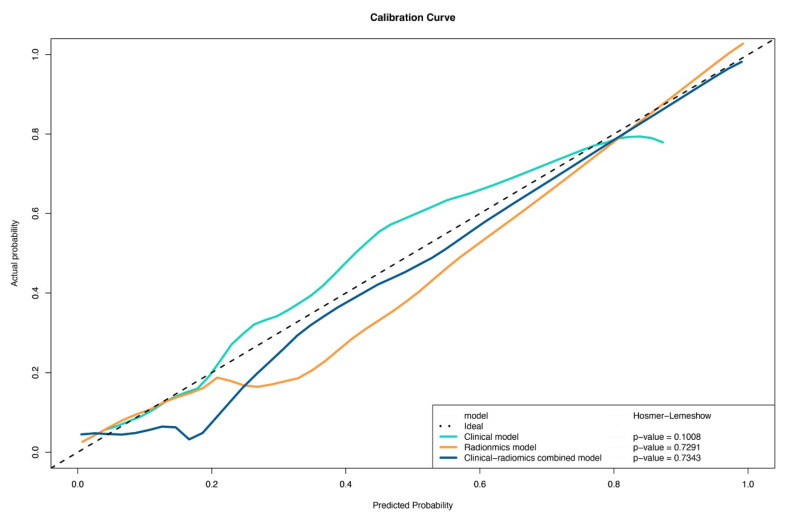
Calibration curve of the radiomics nomogram for predicting PCa in the validation cohort.

**Figure 7 diagnostics-12-03005-f007:**
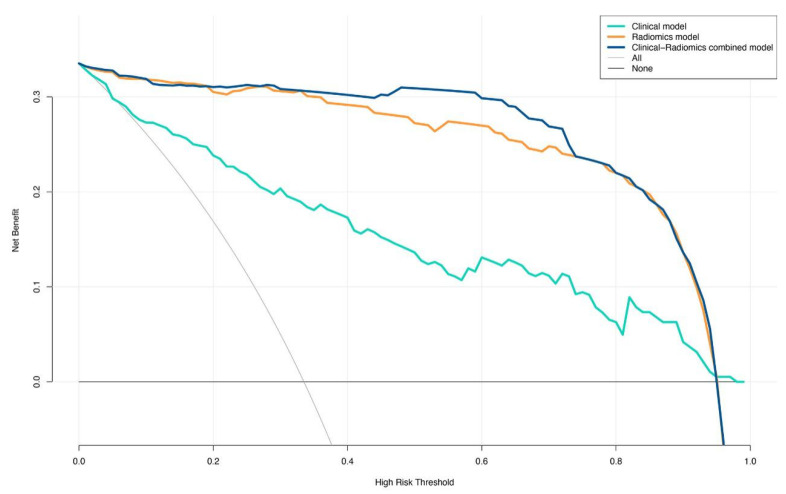
Decision curves of the clinical, radiomics, and clinical-radiomics combined models for predicting PCa in the validation cohort.

**Table 1 diagnostics-12-03005-t001:** The characteristics of all patients.

Characteristics	PCa (*n* = 90)	Non-PCa (*n* = 174)	*p*	Training Cohort (*n* = 191)	Validation Cohort (*n* = 83)	*p*
**Age (y)**	75.0 [53.0, 98.0]	69.5 [47.0, 93.0]	0.015	71.0 [47.0, 98.0]	74.0 [57.0, 93.0]	0.360
**Volume (mL)**	39.0 [14.5, 138]	60.5 [2.70, 432]	<0.001	54.1 [14.2, 432]	53.6 [2.70, 186]	0.484
**tPSA**	7.66 [4.00, 10.7]	6.71 [3.26, 9.95]	0.005	6.98 [3.26, 10.7]	7.35 [4.21, 9.95]	0.749
**fPSA**	1.19 [0.01, 6.06]	1.18 [0.10, 5.13]	0.979	1.20 [0.01, 6.06]	1.14 [0.20, 3.12]	0.512
**f/tPSA**	0.161 [0.0025, 0.652]	0.180 [0.025, 0.523]	0.157	0.177 [0.0025, 0.652]	0.167 [0.0274, 0.382]	0.339
**PSAD**	0.193 [0.0474, 0.513]	0.113 [0.0095, 2.93]	0.037	0.129 [0.0095, 0.513]	0.129 [0.0095, 0.513]	0.173
**PI-RADS v2.1 score**			<0.001			0.993
1	5.00 (5.6%)	52.0 (28.3%)		41.0 (21.5%)	16.0 (19.3%)	
2	7.00 (7.8%)	82.0 (44.6%)		58.0 (30.4%)	31.0 (37.3%)	
3	13.0 (14.4%)	29.0 (15.8%)		30.0 (15.7%)	12.0 (14.5%)	
4	39.0 (43.3%)	19.0 (10.3%)		41.0 (21.5%)	17.0 (20.5%)	
5	26.0 (28.9%)	2.00 (1.1%)		21.0 (11.0%)	7.00 (8.4%)	

**Table 2 diagnostics-12-03005-t002:** Univariate and multiple logistic analysis results of clinical factors.

Predictor	Univariate Analysis			Multiple Analysis		
	β	OR (95%CI)	*p* Value	β	OR (95%CI)	*p* Value
**Age (y)**	0.0748	2.455 [1.501, 4.015]	<0.001	0.0774	2.533 [1.378, 4.656]	0.001
**Volume (mL)**	−0.0186	0.486 [0.308, 0.768]	0.002	NA	NA	0.486
**tPSA**	0.2307	1.871 [1.445, 3.058]	0.013	NA	NA	0.225
**fPSA**	0.0283	1.012 [0.749, 1.393]	0.893	NA	NA	NA
**f/tPSA**	−2.0633	0.803 [0.552, 1.165]	0.247	NA	NA	NA
**PSAD**	8.1946	2.956 [1.568, 3.362]	<0.001	NA	NA	0.714
**PI-RADS v2.1 score**	1.2241	11.569 [5.811, 23.032]	<0.001	1.2439	12.034 [5.847, 24.77]	<0.001

OR, odds ratio; CI, confidence interval; NA, Only when univariate and multivariate regression variable values have statistical significance, that is, when the *p*-value is less than 0.05, corresponding β value and OR value can be obtained. The remaining spaces have no corresponding values and are represented by Not available (NA); PSA, Prostate-specific antigen.

**Table 3 diagnostics-12-03005-t003:** Diagnostic performance of three models in training and validation sets.

	Training Cohort			Validation Cohort		
	Clinical Model	Radiomics Model	Clinical-Radiomics Combined Model	Clinical Model	Radiomics Model	Clinical-Radiomics Combined Model
**AUC (95%CI)**	0.868 [0.813, 0.922]	0.982 [0.964, 0.999]	0.984 [0.968, 1.000]	0.866 [0.783, 0.950]	0.941 [0.888, 0.995]	0.953 [0.907, 0.999]
**Sensitivity (95%CI)**	0.750 [0.630–0.841]	0.953 [0.865, 0.985]	0.953 [0.865, 0.985]	0.846 [0.655, 0.941]	0.808 [0.613, 0.918]	0.885 [0.697, 0.962]
**Specificity (95%CI)**	0.827 [0.751–0.883]	0.961 [0.909, 0.984]	0.984 [0.939, 0.996]	0.842 [0.724, 0.916]	0.965 [0.870,0.991]	0.930 [0.827, 0.973]
**PPV**	0.686 [0.562, 0.789]	0.924 [0.825, 0.972]	0.968 [0.880, 0.994]	0.710 [0.518, 0.851]	0.913 [0.705, 0.985]	0.852 [0.654, 0.951]
**NPV**	0.868 [0.791, 0.920]	0.976 [0.926, 0.994]	0.977 [0.928, 0.994]	0.923 [0.806, 0.975]	0.917 [0.809, 0.969]	0.946 [0.842, 0.986]
**LR+**	4.330 [2.886, 6.494]	24.209 [10.236, 57.257]	60.523 [15.287, 239.619]	5.359 [2.878, 9.977]	23.019 [5.825, 90.973]	12.606 [4.850, 32.762]
**LR−**	0.302 [0.197, 0.464]	0.049 [0.016, 0.147]	0.048 [0.016, 0.144]	0.183 [0.073, 0.453]	0.199 [0.091, 0.439]	0.124 [0.043, 0.361]
**Z**	−4.391	−0.952	NA	−2.154	−1.1227	NA
***p* value**	<0.001	0.341	NA	0.031	0.262	NA

## Data Availability

The patient data used to support the findings of this study are restricted by the Institutional Review Board of Shaanxi Provincial People’s Hospital in order to protect patient privacy.

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
