# Peer review of "MRI-Based Radiomics Nomogram for Predicting Prostate Cancer with Gray-Zone Prostate-Specific Antigen Levels to Reduce Unnecessary Biopsies"

_diagnostics, 2022, doi:10.3390/diagnostics12123005_

Round 1
Reviewer 1 Report
The authors present a very interesting and original paper demonstrating the added value of a combined clinical - radiomics model compared to clinical features alone in predicting prostate cancer in selected set of patients, represented by those with gray-zone PSA levels (4-10 ng/ml). The paper is globally well written with a rigorous statistics, but there is some equivocal sentences in the material and methods section and some discrepancies between this section and results that make not clear the methodology. These flaws should be improved to make the paper suitable for publication.
Major issue
Table 3: The training and validation test was applied for all the three models but this is not declared in the material and methods section.
Page 2, line 81: "PCa or information or prostatic hyperplasia": the sentence is not clear; please reformat
Figure 1: Non PCa are 174 in Table and 184 in material and methods; please solve this discrepancy
Page 3, line 102: what about targeted biopsies? were not they performed at all? In this case the results might have been affected by understimation of Gleason Score and with the diagnosis of PCa until 17,4% as demonstrated by Fiorentino V et al, Histopathological Ratios to Predict Gleason Score Agreement between Biopsy and Radical Prostatectomy. Diagnostics 2020 doi: 10.3390/diagnostics11010010. Anyway, if the length of specimens was >12 mm this understimation may have been avoided. Please, discuss it and add the reference above. Moreover, in case no targeted biopsies were not performed or the length of specimens was <12 mm, or if you are not able to assess these aspects, add this as a limit of the study.
Page 3, line 104: did you check the inter-oberver variability?
Page 4, line 120: what do you mean for radiologic-histologic correlation? did you overlap or fuse the image? Please, explain better
Page 4, line 124: what do you mean for "matched"? Did you perform any other analyses after this matching?
Page 4, line 130: why didn't you analyze DCE imaging?
Page 4, lines 131-132: the sentence is not clear; please reformat
Minor issues:
Provide the values of gray-zone PSA also in the abstract
Specify PI-RADS v2.1 throughout the manuscript
Page 4, line 120: explain the acronymus VOI
MRI protocol: T2-w imaging was acquired in a single plane? Planes of other sequences are not reported at all. Please, add this information and relative parameters to the text or to the Table S1
Page 5, lines 180-182: why malignant lesions were found more in the transitional zone (199) than in the peripheral zone (75)? It should be the contrary as expected from literature. Discuss it.
Table 2: explain in the caption or in the text why the results were "NA" where it appears
Figure 6: explain the acronymus LNM in the caption
Fill the fields at the end of the mansucript: Funding, Institutional Review Board Statement, Informed Consent Statement, Conflicts of Interest
Author Response
Dear Reviewer,
We feel great thanks for your professional review work on our article. As you are concerned, there are several problems that need to be addressed. According to your nice suggestions, we have made extensive corrections to our previous draft, the detailed corrections are listed below.
Point-by-point responses to Reviewers' comments
Reviewer 1
Major issue
1.Table 3: The training and validation test was applied for all the three models but this is not declared in the material and methods section.
Response:Thank you for pointing this out. We have added the suggested content to the manuscript on Statistical analysis section in the material and methods part (Page5, Line 200).
- Page 2, line 81: "PCa or information or prostatic hyperplasia": the sentence is not clear; please reformat
Response:Thanks for your suggestion. We feel sorry for our poor writings. The”PCa or information or prostatic hyperplasia;”has been corrected on” PCa, inflammation or prostatic hyperplasia”(Page 2, Line 83). Thank you for your reminder.
- Figure 1: Non PCa are 174 in Table and 184 in material and methods; please solve this discrepancy
Response:We were really sorry for our careless mistakes.We have corrected Figure 1 and we also feel great thanks for your point out.((Page 3, Figure 1).
- Page 3, line 102: what about targeted biopsies? were not they performed at all? In this case the results might have been affected by understimation of Gleason Score and with the diagnosis of PCa until 17,4% as demonstrated by Fiorentino V et al, Histopathological Ratios to Predict Gleason Score Agreement between Biopsy and Radical Prostatectomy. Diagnostics 2020 doi:3390/diagnostics11010010. Anyway, if the length of specimens was >12 mm this understimation may have been avoided. Please, discuss it and add the reference above. Moreover, in case no targeted biopsies were not performed or the length of specimens was <12 mm, or if you are not able to assess these aspects, add this as a limit of the study.
Response:Your suggestion really means a lot to us. In our study, benign lesions were evaluated based on a standard 12-core systematic transperineal ultrasound-guided prostate biopsy and most of them, the length of specimens was <12 mm. We agree that this is a potential limitation of the study. We have added this as a limitation(Page 13, Line 367-370). As suggested by the reviewer, we have added the reference to support this idea(Page 15, Line 497-499). We also feel great thanks for your point out.
- Page 3, line 104: did you check the inter-oberver variability?
Response: Thank you for pointing this out. Although we agree that this is an important consideration, in our study, two experienced genitourinary pathologists recorded the location and Gleason score, with all disagreements being resolved by consensus. In future study, we will focus on the issue of the inter-oberver variability of pathologists and improve the rigorous of the research.Thank you for your reminder.
- Page 4, line 120: what do you mean for radiologic-histologic correlation? did you overlap or fuse the image? Please, explain better
Response: We agree with the reviewer’s assessment. Our original writing did cause ambiguity, now we modified it according to the reviewer's opinion as follows: Volumes of interest (VOI) was delineated the visible lesion slice by slice based on T2WI and ADC map. The location and size of the lesions were determined as follows: (I) the detailed records of the prostate system punctures (injection site and depth) and pathological diagnostic results were used to determine the location and nature of the lesion; (II) the location described by pathology was matched to the corresponding lesion on the MRI images. Accordingly, throughout the manuscript, we have explained the suggested content to the manuscript on Page 3, Line 110-116 and Page 4, Line 138-139,143-147.
- Page 4, line 124: what do you mean for "matched"? Did you perform any other analyses after this matching?
Response: Thank you again for your positive comments and valuable suggestions to improve the quality of our manuscript. The "matched"means PI-RADS V2.1 results, biopsy results, and the radiologist’s segmentation area, which the locations of the lesions described by the three methods are matched. The location and size of the lesions were determined as follows: (I) the detailed records of the prostate system punctures (injection site and depth) and pathological diagnostic results were used to determine the location and nature of the lesion; (II) the location described by pathology was matched to the corresponding lesion on the MRI images; and (III) combined with the PI-RADS V2.1 scoring diagnostic criteria, the range of lesions was determined.(Page 3, Line 110-116; Page 4, Line 143-147)
(Li M, Chen T, Zhao W, et al. Radiomics prediction model for the improved diagnosis of clinically significant prostate cancer on biparametric MRI. Quant Imaging Med Surg. 2020 Feb;10(2):368-379. doi: 10.21037/qims.2019.12.06.)
- Page 4, line 130: why didn't you analyze DCE imaging?
Response:Thank you for your nice comments on our article. T2WI and ADC images allow an accurate detection and localization of suspicious PCa, in addition to reduction of the time required to complete the study, lower costs without using gadolinium. Moreover, potential risks related to the use of gadolinium-based contrast media such as nephrogenic systemic fibrosis, renal failure and brain accumulation of gadolinium would be reduced. In general, T2WI and ADC are the most common sequences selected by investigators for radiomics research.Therefore, we chose to derive radiomics features from T2WI and ADC only for this study. In addition, using too many sequences will affect its clinical application because of time-consuming and laborious image segmentation, so it is more important to choose the valuable sequence. Therefore, we conclude that bp-MRI–based radiomic may contribute to differentiate Pca from non PCa and to risk stratification without the need for additional MRI sequences such as DCE.
References:
[1]Niu XK,Chen ZF,Chen L, et al. Clinical Application of Biparametric MRI Texture Analysis for Detection and Evaluation of High-Grade Prostate Cancer in Zone-Specific Regions.AJR Am J Roentgenol 2018; 210: 549-556.
[2]Scialpi M,D'Andrea A,Martorana E, et al. Biparametric MRI of the prostate.Turk J Urol 2017; 43: 401-409.
[3]McDonald RJ,McDonald JS,Kallmes DF, et al. Intracranial Gadolinium Deposition after Contrast-enhanced MR Imaging.Radiology 2015; 275: 772-82.
[4]Khalvati F,Wong A,Haider MA. Automated prostate cancer detection via comprehensive multi-parametric magnetic resonance imaging texture feature models.BMC Med Imaging 2015; 15: 27.
[5]Kwak JT,Xu S,Wood BJ, et al. Automated prostate cancer detection using T2-weighted and high-b-value diffusion-weighted magnetic resonance imaging.Med Phys 2015; 42: 2368-2378.
9.Page 4, lines 131-132: the sentence is not clear; please reformat
Response:We sincerely thank the reviewer for careful reading. As suggested by the reviewer, we have corrected the “Four types of features (first-order statistics, texture features, shape features, and wavelet filter and Laplacian of Gaussian filter features) for a total of 3146 features were extracted from axial ADC and T2WI sequences.” into “A total of 3146 features of four types (first-order statistics, texture features, shape features, and wavelet filter and Laplacian of Gaussian filter features) were extracted from axial ADC and T2WI sequences”.(Page 4,Line 152)
Minor issues:
- Provide the values of gray-zone PSA also in the abstract
Response:We think this is an excellent suggestion. We have added the values of gray-zone PSA also in the abstract.(Page 1, Line 11-12)
2.Specify PI-RADS v2.1 throughout the manuscript
Response:Thank you for this suggestion. We have added the suggested content to the manuscript as follows:( Page4, Line126-131).
“For the transition zone (TZ) lesion, T2WI remains the key sequence, with using of DWI features to upgrade atypical nodules to category 3 lesions the for staging lesions. For peripheral zone(PZ)lesion, DWI remains the key sequence, with using DCE for DWI score 3 lesions. The radiologists independently assigned each lesion a score of 1–5 for T2WI, a score of 1–5 for DWI, “+”or “-” for DCE,and an overall PI-RADS assessment category according to PI-RADS v2.1.”
.
3.Page 4, line 120: explain the acronymus VOI
Response:Thank you for pointing this out. As suggested by the reviewer, we have added the full name volumes of interest (VOI). (Page4, Line 138).
4.MRI protocol: T2-w imaging was acquired in a single plane? Planes of other sequences are not reported at all. Please, add this information and relative parameters to the text or to the Table S1
Response:We agree with the reviewer’s assessment. Accordingly, throughout the manuscript, we have added the information and relative parameters to the table S1.
5.Page 5, lines 180-182: why malignant lesions were found more in the transitional zone (199) than in the peripheral zone (75)? It should be the contrary as expected from literature. Discuss it.
Response:Thank you for pointing this out. We feel sorry for our poor writings. However, a total of 274 patient were included in this study, of which 90 patients had PCa and 184 patients had benign lesions. The most of benign lesions(177) were benign prostatic hyperplasia which were located in the transitional zone. Therefore, 75 lesions were found to have originated from the peripheral zone, while 199 lesions were located in the transitional zone.
Our original writing did cause ambiguity, now we modified it according to the reviewer's opinion as follow “Among them, 177 lesions were benign prostatic hyperplasia which were located in the TZ.” (Page 6, Line225-226).
6.Table 2: explain in the caption or in the text why the results were "NA" where it appears
Response:Thank you for pointing this out. Our original writing did cause ambiguity, now we explained it according to the reviewer's opinion as follow: Only when univariate and multivariate regression variable values have statistical significance, that is, when P value is less than 0.05, corresponding β value and OR value can be obtained. The remaining spaces have no corresponding values and are represented by Not available(NA).(Page 7,Table 2)
- Figure 6: explain the acronymus LNM in the caption
Response:We feel sorry for our carelessness. In our resubmitted manuscript, the typo is revised. Thanks for your correction. We have corrected the “LNM” into “PCa”.(Page 11, Figure 6)
- Fill the fields at the end of the mansucript: Funding, Institutional Review Board Statement, Informed Consent Statement, Conflicts of Interest
Response:As suggested by the reviewer, we have added the above contents. (Page13)
Funding: This work was supported by the Support Plan for Scientific and Technological Talents of Shaanxi Provincial People's Hospital (grant number 2021JY-43).
Institutional Review Board Statement: The study was approved by The institutional review board ethics committee of Shaanxi Provincial People's Hospital.
Informed Consent Statement: Because this was a retrospective study based only on
medical records and all data were analyzed in our hospital, patient consent was waived.
Conflicts of Interest: The authors declare no conflict of interest.
We tried our best to improve the manuscript and made some changes in the manuscript. These changes will not influence the content and framework of the paper. And revised portion are marked with tracked changes in the paper. We appreciate for reviewer’ warm work earnestly, and hope that the correction will meet with approval.
Once again, thank you very much for your comments and suggestions.
Yours sincerely,
Li Zhang

Reviewer 2 Report
Dear Authors,
Thanks for this interesting work.
The purpose is relevant and the study seems to be well conducted but some details have to be add in your results:
- Flowchart was unclear : you included 976 patient L 76-78 but it is different in flow chart.
- L 180-182: Most lesions are located in the transitional zone which is not common. Please explain.
- Please describe your population:
How many people have prostate cancer in each group (training and validation set)?
Detail the diagnostic modalities: How many have biopsies? How many have surgical specimen?
Other: L44: change “stones” by “prostate”
Author Response
Dear Reviewer,
We feel great thanks for your professional review work on our article. As you are concerned, there are several problems that need to be addressed. According to your nice suggestions, we have made extensive corrections to our previous draft, the detailed corrections are listed below.
Point-by-point responses to Reviewers' comments
Reviewer2
1.Flowchart was unclear : you included 976 patient L 76-78 but it is different in flow chart.
Response:We feel sorry for our carelessness. In our resubmitted manuscript, the typo is revised. Thanks for your correction(Page 2, Line78).
2.- L 180-182: Most lesions are located in the transitional zone which is not common. Please explain.
Response:Thank you for pointing this out. We feel sorry for our poor writings. However, a total of 274 patient were included in this study, of which 90 patients had PCa and 184 patients had benign lesions. The most of benign lesions(177) were benign prostatic hyperplasia which were located in the transitional zone. Therefore, 75 lesions were found to have originated from the peripheral zone, while 199 lesions were located in the transitional zone.
Our original writing did cause ambiguity, now we modified it according to the reviewer's opinion as follow “Among them, 177 lesions were benign prostatic hyperplasia which were located in the TZ.” (Page 6, Line225-226).
3.- Please describe your population:
How many people have prostate cancer in each group (training and validation set)?
Response:As suggested by the reviewer, we have supplemented the results accordingly. “A total of 64(33.5%) of 191 patients in the training cohort and 26 (31.2%) of 83 patients in the validation cohort were diagnosed with PCa.” (Page 6, Line 226-227)
4.Detail the diagnostic modalities: How many have biopsies? How many have surgical specimen?
Response:We think this is an excellent suggestion. We have supplemented the results accordingly. “As for the reference standard, 184 patients with benign lesions and 41 patients with PCa did not undergo radical prostatectomy, and the pathological results were determined by TRUS-guided biopsy. Other 49 patients with PCa did undergo radical prostatectomy with surgical specimen.”(Page 6, Line 227-231.)
5.Other: L44: change “stones” by “prostate”
Response:We sincerely thank the reviewer for careful reading. As suggested by the reviewer, we have corrected the “stones” into “prostate”.(Page2, Line46)
We tried our best to improve the manuscript and made some changes in the manuscript. These changes will not influence the content and framework of the paper. And revised portion are marked with tracked changes in the paper. We appreciate for reviewer’ warm work earnestly, and hope that the correction will meet with approval.
Once again, thank you very much for your comments and suggestions.
Yours sincerely,
Li Zhang
Round 2
Reviewer 1 Report
The paper has been improved significantly by te authors, great job. Anyway, some other few minor corrections are needed.
I still see “information” instead of inflammation in the section 2.1.
I agree with the conclusion that bp-MRI–based radiomic may contribute to differentiate Pca from non PCa and to risk stratification, but DCE is still needed to give a conclusive PIRADS score. Potential risks related to the use of gadolinium-based contrast media (nephrogenic systemic fibrosis, renal failure and brain accumulation of gadolinium), as the longer duration and higher costs due to the use of gadolinium do not justify the elimination of DCE from the MRI protocol. Please add this as a limitation of the study.
Explain the acronyms T2WI and T1WI as T2-weighted images and T2-weighted images in the section 2.2.
Author Response
Dear Editor and Reviewer,
Thank you again for your positive comments on our manuscript.We have carefully reviewed the comments and have revised the manuscript accordingly. Our responses are given in a point-by-point manner below. Changes to the manuscript are shown in red.
1.I still see “information” instead of inflammation in the section
Response:We were really sorry for our careless mistakes. Thank you for your reminder. We have corrected the “information” into “inflammation”.(Page 2, Line 81)
2. I agree with the conclusion that bp-MRI–based radiomic may contribute to differentiate Pca from non PCa and to risk stratification, but DCE is still needed to give a conclusive PIRADS score. Potential risks related to the use of gadolinium-based contrast media (nephrogenic systemic fibrosis, renal failure and brain accumulation of gadolinium), as the longer duration and higher costs due to the use of gadolinium do not justify the elimination of DCE from the MRI protocol. Please add this as a limitation of the study.
Response:Thank you for this suggestion. We agree that this is a potential limitation of the study. We have added this as a limitation on Page 13, Line 343-346.
3. Explain the acronyms T2WI and T1WI as T2-weighted images and T2-weighted images in the section 2.2.
Response:We sincerely thank the reviewer for careful reading. We have corrected them and we also feel great thanks for your point out.(Page 3, Line 92-93)
We tried our best to improve the manuscript and made some changes in the manuscript. We appreciate for reviewer’ warm work earnestly, and hope that the correction will meet with approval.
Once again, thank you very much for your comments and suggestions.
Yours sincerely,
Li Zhang